# Hyperthermia Treatment Monitoring via Deep Learning Enhanced Microwave Imaging: A Numerical Assessment

**DOI:** 10.3390/cancers15061717

**Published:** 2023-03-11

**Authors:** Álvaro Yago Ruiz, Marta Cavagnaro, Lorenzo Crocco

**Affiliations:** 1CNR-IREA National Research Council of Italy, Institute for Electromagnetic Sensing of the Environment, 80124 Naples, Italy; 2Department of Information Engineering, Electronics, and Telecommunications, University of Rome “La Sapienza”, 00184 Rome, Italy

**Keywords:** deep learning, microwave imaging, hyperthermia treatment, temperature monitoring

## Abstract

**Simple Summary:**

Non-invasive temperature monitoring during hyperthermia cancer treatment is of paramount importance. It allows physicians to verify the therapeutic temperature is reached in the treated area. Currently, only superficial or invasive thermometry is performed on a clinical level. Magnetic resonance thermometry has been proposed as a a non-invasive alternative but its applicability is limited. Conversely, microwave imaging based thermometry is a potential low cost candidate for non-invasive temperature monitoring. This works presents a computational study in which the use of deep learning is proposed to face the challenges related to the use of microwave imaging in hyperthermia monitoring.

**Abstract:**

The paper deals with the problem of monitoring temperature during hyperthermia treatments in the whole domain of interest. In particular, a physics-assisted deep learning computational framework is proposed to provide an objective assessment of the temperature in the target tissue to be treated and in the healthy one to be preserved, based on the measurements performed by a microwave imaging device. The proposed concept is assessed in-silico for the case of neck tumors achieving an accuracy above 90%. The paper results show the potential of the proposed approach and support further studies aimed at its experimental validation.

## 1. Introduction

Hyperthermia treatment (HT) is an adjunctive cancer therapy in which cancerous tissues are heated up to 40–43 °C to help damage malignant cells. This is achieved by sensitizing the cells to radiation in radiotherapy (RT) or increasing the permeability of the cell membrane to drugs in chemotherapy [1]. The HT synergistic effect inhibits DNA damage repair and increases blood flow (thereby enhancing oxygenation). In particular, RT is more cytotoxic in normal oxygenation conditions than in hypoxic environments. In this regard, HT may act as a complementary treatment of RT in cases of tumor hypoxia by sensitizing the cells to heat in low-oxygen conditions [2]. Additionally, HT induces the activation of heat shock proteins and a tumor-specific immune response.

Local, regional, and whole-body HT may be considered depending on the size and location of the treatment area. HT consists of a heating applicator that is able to supply heat to the treatment area. However, heat transfer is not easily applied in a localized manner. Tracking the temperature during the HT is needed to verify that the therapeutic temperature is reached in the target tissue and that the surrounding healthy tissue is not heated (and, thus, not damaged) giving rise to the so-called hot spots. In clinical practice, this task is performed by means of superficial thermometry or by directly inserting thermal sensors into the tissue, such as thermocouples or optical fibers. However, these approaches are limited to superficial HT or are invasive (due to the need for inserting the probes), respectively. Furthermore, since probes are sensitive to temperature in the insertion location, invasive thermometry is limited to local measurements. Patient feedback is also taken into consideration during HT, but it is subject to individual patient perception.

Considering the above reasons, there is increasing interest in the development of fully non-invasive temperature monitoring approaches capable of overcoming the above issues. To this end, magnetic resonance (MR) thermometry [3] has been proposed based on temperature-sensitive parameters involved in MR, such as proton resonance frequency, diffusion coefficient, $T_{1}$ and $T_{2}$ relaxation times, magnetization transfer, and proton density. However, MR thermometry is only feasible in 15% of the treatments, mainly when the treatment area is in the limbs or the HT device fits inside the bore of the MR system [4,5]. Moreover, HT-MR has a high cost and its outcomes are affected by the inaccuracies introduced by movements (e.g., breathing, bowels, blood vessels). Finally, standard HT applicators may not be suitable for operation in the presence of electromagnetic (EM) fields generated by an MR machine due to EM compatibility issues, and thus, they must be redesigned accordingly.

Microwave imaging (MWI) [6] is an emerging medical imaging modality, which may represent a potential candidate technology for non-invasive temperature monitoring in HT. MWI is based on the scattering phenomenon that occurs when an EM field interacts with an object. Such an interaction perturbs the field according to the morphological properties of the target and its EM properties, i.e., dielectric permittivity and electric conductivity. By processing the perturbed field, known as the interrogating one, it is possible to build an image of the investigated scenario in terms of a map of the spatial distribution of the EM properties in the region under test. In the field of medical imaging, MWI is applicable thanks to the different EM properties that characterize human tissues, as well as the different statuses (healthy vs. pathological) of certain tissues, due to the non-ionizing nature and low power of the EM fields involved, as well as the low cost of the underlying components, MWI has garnered significant research attention in recent years, particularly in the development of devices for the early diagnosis of breast cancer [7] and cerebrovascular diseases [8].

As there is a known relationship between temperature and tissue EM properties [9], MWI can also be used to monitor temperature during HT. This was suggested by Bolomey and colleagues many years ago [10]. Notably, unlike MR thermometry, MWI-based thermometry would rely on portable, cost-effective devices, and would not pose significant issues in terms of EM compatibility when integrated with HT systems. On the other hand, the mathematical nature of the problem underlying MWI poses a non-trivial challenge in the practical implementation of the technique. MWI involves solving an inverse scattering problem (ISP) [6,11], which is known to be nonlinear and ill-posed, leading to instabilities in the solution process. As such, inaccurate estimations of the EM properties (and, hence, their temperature) may negatively impact the reliability of the treatment monitoring technique. For this reason, while the use of MWI to monitor the temperature in HT has been investigated by several authors [12,13,14,15], the possibility of quantitatively sensing and tracking the changes in the EM properties occurring due to thermal treatment via MWI is still an open issue.

Recently, there has been increasing attention given to the use of machine learning tools to address the difficulties associated with solving the ISP in MWI [16]. In particular, the ability of deep learning (DL) algorithms to solve complex nonlinear regression problems has been explored, especially in the development of physics-driven solution frameworks. In the latter, DL architectures are combined with standard MWI approaches that supply the neural network with domain knowledge that either makes the training of the network easier or its predictions more efficient [17,18]. However, to the best of our knowledge, the use of these powerful tools to provide a robust solution to ISP in the framework of temperature monitoring via MWI has never been explored.

According to the above, this paper describes a physics-driven DL framework to perform reliable, quantitative, and robust MWI-based monitoring and tracking of the temperature evolution during HT. The basic assumption of the proposed approach is that accurate knowledge of the treated region (in terms of tissue segmentation) is available thanks to the treatment planning stage routinely performed before the actual treatment. Prior knowledge about the changes in EM properties within the temperature range of HT allows for the ISP to be formulated as a linear inverse problem, with remarkable simplification. However, the problem remains ill-posed and may result in inaccurate or unstable estimations, which can be addressed by exploiting DL. In particular, the proposed approach consists of a first stage in which the MWI raw data are processed by means of a linearized inversion algorithm based on the distorted Born approximation [6], in which the stability of the solution is supplied by adopting the truncated singular value decomposition [19]. This step focuses on an (approximated) estimation of the EM properties in the regions of interest, i.e., the tumor to be treated and some surrounding regions to be protected. In the second step, the MWI imaging results are supplied to a convolutional neural network (CNN), trained in such a way as to classify the incoming results according to their temperature. The output of the CNN is a set of classification labels, which provides immediate information on the heated or unheated status of the region of interest.

In the following sections, we provide a detailed description of the proposed approach and present a simulated example of monitoring an HT treatment of neck tumors to provide an initial proof of concept.

## 2. Material and Methods

### 2.1. Electromagnetic Properties of Tissue

The goal of a general MWI system consists of the characterization of the target in terms of its morphological features, such as shape and position, and its electromagnetic properties, namely, dielectric permittivity, electric conductivity, and magnetic permeability.

Some materials, including biological tissues, are non-magnetic, meaning they have free space permeability $\mu =\mu_{0}=4\pi \times 10^{-7}$ H/m or relative permeability $\mu_{r}=\mu /\mu_{0}=1$. Consequently, permittivity and conductivity are usually the quantities of interest. Exploiting the formulation for time-harmonic waves [20], they can be conveniently incorporated into the relative complex permittivity, defined as
(1)$$\varepsilon_{r}=\varepsilon_{r}^{’}-j\varepsilon_{r}^{’’},$$
where *j* stands for the imaginary unit.

In (Equation 1), the real and imaginary parts of $\varepsilon_{r}$ encode dielectric and conductive phenomena depending on the specific features of the media at hand. Biological tissues present dispersive behaviors, meaning that their EM properties depend on the operating frequency *f*. Moreover, and more important to our purposes here, their EM properties exhibit a dependence on temperature [9], which lays the basis for the feasibility of MWI-based monitoring of temperature in HT treatments.

Accordingly, the EM behavior of tissues may be modeled using a temperature dependent Cole–Cole model [9]:(2)$$\varepsilon_{r}\left(\omega ,T\right)=\varepsilon_{\infty }\left(T\right)+\frac{\Delta \varepsilon_{1}\left(T\right)}{1+{\left(j\omega \tau_{1}\left(T\right)\right)}^{1-\alpha_{1}}}+\frac{\Delta \varepsilon_{2}\left(T\right)}{1+{\left(j\omega \tau_{2}\left(T\right)\right)}^{1-\alpha_{2}}}+\frac{\sigma \left(T\right)}{j\omega \varepsilon_{0}}$$
where ω=2πf is the angular frequency of the time-harmonic field, *T* is the temperature, $\varepsilon_{\infty }$ is the permittivity at very high frequencies, $\Delta \varepsilon_{1}$ and $\Delta \varepsilon_{2}$ are the dispersion amplitudes with their corresponding relaxation times $\tau_{1}$, $\tau_{2}$, and σ is the conductivity, measured in [S/m]. $\alpha_{1}$ and $\alpha_{2}$ are dispersion-broadening parameters and $\varepsilon_{0}=8.858\times 10^{-12}$ F/m is the dielectric permittivity in the vacuum.

### 2.2. The Microwave Imaging Problem

The main challenge in MWI is solving the ISP, which is addressed in this paper as a 2D problem. Specifically, the MWI procedure is applied to a single slice of the body, as depicted in Figure 1. In a typical MWI setting, let Ω denote the selected imaging domain (i.e., the slice under treatment) enclosing a cross-section Σ of a collection of tissues, which are assumed to be elongated bodies that extend to infinity. Such a circumstance implies that the (unknown) properties of the tissues do not change across the longitudinal *z*-axis, ∂/∂z=0 (2D problem). Thus, changes only happen in the transverse plane $\nabla =\nabla_{trans}=\left(x\frac{\partial }{\partial x}+y\frac{\partial }{\partial y}\right)$. Additionally, such tissues are probed with time-harmonic incident fields $E_{inc}$ radiating in a transverse magnetic (TM) mode under the time convention $e^{j\omega t}$. In TM-polarization, the magnetic field is transverse to the z-axis, ($H_{x},H_{y}$), while the electric field only has the longitudinal component, $E_{z}$. As the magnetic and electric fields are uniquely linked through the Maxwell equations for any fixed scenario, in these conditions the ISP can be cast as a scalar problem by considering only the electric field.

The interrogating process begins with the incident fields impinging the imaging domain. Such fields are radiated from a set $v=1,\cdots ,N_{v}$ of transmitting antennas $T_{x}^{\left[v\right]}$, located outside Ω, on a curve Γ enclosing it. For the generic transmitter positioned in $r_{v}\in \Gamma$, the interaction between $E_{inc}$ and the tissues gives rise to a scattered field $E_{S}$, which can be measured by a set $m=1,\cdots ,N_{m}$ of receiving antennas $R_{x}^{\left[m\right]}$, positioned in $r_{m}\in \Gamma$. More precisely, the actual field measurements at $R_{x}^{\left[m\right]}$ correspond to a total field $E_{tot}$, which is a superposition of both incident and scattered fields $E_{tot}=E_{inc}+E_{S}$. The interactions between the interrogating fields and the tissues are governed by the state and data equations: (3)$$\begin{array}{c} E_{tot}\left(r,r_{v}\right)=E_{inc}\left(r,r_{v}\right)+k_{b}^{2}\int_{\Omega }g\left(r,r^{’}\right)\chi \left(r^{’}\right)E_{tot}\left(r^{’},r_{v}\right)dr^{’} \end{array}$$(4)$$\begin{array}{c} E_{S}\left(r_{m},r_{v}\right)=k_{b}^{2}\int_{\Omega }g\left(r_{m},r^{’}\right)\chi \left(r^{’}\right)E_{tot}\left(r^{’},r_{v}\right)dr^{’} \end{array}$$
where for a given angular frequency ω, $k_{b}=\omega \sqrt{\mu_{0}\varepsilon_{0}\varepsilon_{rb}}$ is the wavenumber of the background medium in which the probing antennas are located, whose relative complex permittivity is $\varepsilon_{rb}$. *g* denotes the scalar Green’s function, i.e., the impulsive response in the background medium, and $\chi =\varepsilon_{r}/\varepsilon_{rb}-1$ is the *contrast function*, which encodes the EM properties of the targeted tissues. The retrieval of χ from (Equation 3) and (Equation 4) is the goal of the ISP underlying MWI.

### 2.3. Deep Learning Microwave Imaging Framework for HT Temperature Monitoring

During HT, physicians aim to monitor the temperature to determine whether certain areas are exceeding a defined maximum temperature safety threshold (in addition to patient feedback), while ensuring that the temperature in the tumor is within the therapeutic range and not too hot, which could lead to ablation, and, thus, other protocols should be used [1].

In this respect, the devised DL-MWI monitoring system could support the physician acting as a cooperative safety system which confirms the achievement of the treatment endpoint in the tumor and points out hazardous heating in temperature sensitive regions.

This approach can be naturally turned into a classification problem in which a categorical label is assigned to each region of interest (ROI), i.e., the tumor region or/and the surrounding healthy tissues, according to the temperature. In particular, in the tumor region, the expected outcome reaches the therapeutic temperature, whereas, in the healthy tissue, the goal is to avoid hot spots.

As illustrated in Figure 2, such an application can be conveniently cast in terms of a physics-assisted DL-MWI framework involving the processing of the raw data measured by the MWI device through a suitable MWI algorithm, in order to obtain an image of the treatment area, followed by a deep learning architecture in charge of classifying the incoming images into their corresponding temperature labels. Obviously, some surrounding tissues of the tumor tolerate heat better than others. In fact, hot spots in nervous tissue may be more dangerous to the patient than those happening in other areas given the known heat sensitivity of this type of tissue [1]. For this reason, the proposed framework is conceived not as a general hot spot detection framework but rather as a monitoring framework of the tumor as well as temperature-sensitive ROIs.

#### 2.3.1. Microwave Imaging Processing

The proposed DL-MWI framework uses a MWI algorithm to process the measured raw data and provide a 2D image of the treated region. From this image, the DL architecture derives the temperature status of the tumor and the healthy tissue to be protected. Due to the small perturbations of the EM properties in the treated tissue corresponding to temperature increases ΔT during HT [9], a linear approximation of the EM scattering phenomenon, namely the distorted-wave Born approximation (DWBA) [21], can be utilized.

To introduce the DWBA formulation, let us denote with $t_{0}$ the starting time of the treatment and with $t_{s}$ the generic time instant during the treatment, and with $\chi^{\left[0\right]}$ and $\chi^{\left[s\right]}$ the corresponding contrast functions. The differential contrast
(5)$$\delta \chi =\chi^{\left[s\right]}-\chi^{\left[0\right]}$$
provides information on the variation of the tissue EM properties during the treatment, which in turn reflects the temperature changes within the tissue. As these changes are small, it follows that $\left|\delta \chi |<<\;\right|\chi^{\left[s\right]}|$, $\left|\delta \chi |<<\;\right|\chi^{\left[0\right]}|$. Accordingly, $E_{tot}^{\left[s\right]}\approx E_{tot}^{\left[0\right]}$, where $E_{tot}^{\left[0\right]}$ is the field in the ROI at the starting stage of the treatment, i.e., when all of the tissues are at physiological temperatures and $E_{tot}^{\left[s\right]}$ is the field at $t_{s}$. Notably, considering that HT treatments are preceded by a treatment planning stage that involves using MR followed by accurate segmentation of the tissues in the treatment area [1], it is possible to compute the field $E_{tot}^{\left[0\right]}$ via EM simulations.

For the small perturbation regime underlying the DWBA, and assuming δχ as the unknown of the MWI problem, (Equation 3)–(Equation 4) reduce to a single linear equation, which reads:(6)$$\Delta E_{S}\left(r_{m},r_{v}\right)=k_{b}^{2}\int_{\Omega }E_{tot}^{\left[0\right]}\left(r^{’},r_{m}\right)E_{tot}^{\left[0\right]}\left(r^{’},r_{v}\right)\delta \chi \left(r^{’}\right)dr^{’}=L_{\Omega }\delta \chi$$
where $L_{\Omega }$ is a short notation for the linear and compact integral operator appearing in the intermediate term and $\Delta E_{S}=E_{S}^{\left[s\right]}-E_{S}^{\left[0\right]}$ is the differential scattered field obtained by subtracting the scattered field measured by the MWI system in the unheated conditions, $E_{S}^{\left[0\right]}$, from the one at the time $t_{s}$ of the treatment $E_{S}^{\left[s\right]}$. Note that the role of Green’s function in Equation (Equation 6) is played by $E_{tot}^{\left[0\right]}\left(r^{’},r_{m}\right)$ since the unheated condition is considered as the reference scenario.

The linear ill-posed problem cast via the DWBA Equation (Equation 6) can then be solved in a regularized form via the truncated singular value decomposition (TSVD) [6] to provide a stable reconstruction. The TSVD inversion of Equation (Equation 6) reads:(7)$$\delta \chi =\sum_{p=1}^{P_{cut}}\frac{1}{\lambda_{p}}\left(\nu_{p}u_{p}^{H}\Delta E_{S}\right)$$
where $\lambda_{p}$ stands for the *p*-th singular value of $L_{\Omega }$ and $u_{p},\nu_{p}$ for its left and right *p*-th singular vectors, respectively. $P_{cut}$ is the regularization parameter that truncates the summation in (Equation 7). It is worth noting that the estimation of the differential contrast provided by (Equation 7) can be achieved in real time since its computationally intensive part (the evaluation of the SVD of $L_{\Omega }$) is done offline before the processing of the measured data.

The above DWBA inverse formula refers to the evaluation of the differential contrast over the entire treated region Ω. However, the dimensions and positions of the tumor and the healthy tissue to be preserved are a priori known from the treatment planning stage. Hence, it can be convenient to restrict the ISP to the sole identification of the contrast perturbations at those locations, say $\Omega_{1}$ and $\Omega_{2}$, which are the ones of actual clinical interest. In doing so, a further simplification is introduced by the fact that, since the DWBA holds, the scattering phenomenon is localized and the mutual interaction between the two ROIs are negligible. Hence, the imaging of the two ROIs can be handled separately by computing the TSVD images for the operators $L_{\Omega_{1}}$ and $L_{\Omega_{2}}$, obtained by restricting $L_{\Omega }$ to the relevant ROIs. The resulting estimated differential contrasts $\delta \chi_{1}$ and $\delta \chi_{2}$ represent the input of the DL architecture.

#### 2.3.2. Deep Learning Architecture for Classification

Once the MWI processing has generated the two images of the tumor and the healthy tissue to be protected, two independent DL architectures can be trained—one in charge of monitoring the temperature in $\Omega_{1}$ and one in charge of monitoring the temperature in $\Omega_{2}$. Among the possible choices, here, two convolutional neural networks (CNNs), both with identical structures, are used. CNNs are broadly applied in classification tasks, given their superior performances as compared to traditional algorithms. Their most important characteristic involves the use of concatenated convolutional layers, where *K* kernels are convoluted over incoming images to provide a solution [22]. The internal structure of one of the two CNNs is shown in Figure 3.

The heating statuses of the two ROIs can be presented as categorical labels derived from CNN classifiers. However, there is a difference between the expected outcomes of the two cases. The tumor is expected to stay in the therapeutic regime between two thresholds; therefore, the tumor CNN has to work with $N_{c}=3$ classes: {unheated, therapeutic, hot}. The healthy tissue must remain unheated during the whole treatment. Therefore, $N_{c}=2$ classes are required to handle its case: {unheated, hot}.

The parameters W of the two CNNs must be optimized based on a loss function. For categorical labels, the categorical cross-entropy is typically used:(8)$$L_{e}\left(W\right)=-\frac{1}{NN_{c}}\sum_{n=1}^{N}\sum_{c=1}^{N_{c}}C_{c}^{\left[n\right]}\cdot log\;\;{\hat{C}}_{c}^{\left[n\right]}\left(W\right)$$
where *N* corresponds to the dataset size, C is the ground truth label that should be retrieved and only supplied during the training, and $\hat{C}$ is the output prediction.

Since hot spots may appear in any of the tissues surrounding the tumor, monitoring more than two regions could be needed. Notably, this does not require substantial changes to the proposed approach. More precisely, the overall DL-MWI framework would be the same, while only the number of TSVD images and CNNs would have to be changed. Accordingly, without loss of generality, the following proof of concept is discussed for the case of two regions. However, the proposed approach is not limited to such regions.

### 2.4. Assessment of the DL-MWI Framework for Temperature Monitoring in Neck Tumor Hyperthermia

In the following, the above-described DL-MWI framework is developed and numerically assessed for the case of neck tumors. More precisely, in this case, the objective is the simultaneous monitoring of the tumor and the spinal cord heating status. A scheme of the whole framework is illustrated in Figure 4.

#### 2.4.1. Anatomical Model

To perform the simulated study aimed at assessing the proposed approach, a numerical phantom of the neck was built. According to the assumed 2D geometry, a neck slice was extracted from the Ella anatomical model [23], which has been broadly adopted for numerical studies involving interactions between EM waves and the human body. Then, a target with a circular cross-section and a radius of 4.5 mm (with the EM properties of a nodular tumor [24]) was added to the thyroid gland to simulate the pathological condition. The tissue segmentation and electromagnetic properties of the resulting phantom are shown in Figure 5. The EM behaviors of all tissues were modeled using the Cole–Cole model Equation (Equation 2) with the parameters obtained from a publicly available database [25]. However, for the pathological tissues, the parameters were taken as those of a nodular goiter tumor [26].

#### 2.4.2. MWI Simulations

To simulate the measurements generating the MWI raw data, the interactions between the phantom described above and the MWI antennas were computed by numerically solving the scattering Equations (Equation 3) and (Equation 4). To this end, proprietary software based on the method of moments (MoM) [27,28] and the conjugate gradient fast Fourier transform (CG-FFT) [29] was exploited.

In the simulations, the antennas used to generate the MWI data are modeled as point sources and their arrangements around the neck are depicted on the left side of Figure 4. As can be noted, the $N_{v}=N_{m}=12$ antennas are not evenly spaced, so there are gaps to accommodate the antennas of the HT applicator and those of the MWI system. In particular, assuming that the HT applicator is made of three antennas, one at the front of the neck and the other two at its sides, the MWI antennas are positioned in groups of three at each side of the HT antenna placed in front of the tumor, whereas the remaining six were positioned behind the neck. In order to compensate for the limited number of measurements, frequency diversity was exploited by simulating the measurements of the data at $N_{f}=10$ frequencies evenly spaced in the f=[0.9,1.0,⋯,1.8] GHz range. Such range of frequencies, as well as the EM properties of the water mixture used as the background medium were chosen in accordance with previous studies [30].

To simulate the changes in the tissue EM properties during the treatment, the MWI simulations were repeated for several conditions. For the tumor, 10 different values of the EM properties were considered, corresponding to the temperatures in the T= [37–46] °C range, whereas for the spinal cord, the selected temperature range was narrower, T= [37–40] °C, as the goal was to detect the switch between two conditions (heated–unheated). For both tissues, the Cole–Cole parameters were modified according to the model in [9]. The differential data $\Delta E_{S}$ and the MWI raw data were computed for each of the considered temperature conditions, and the scattered fields resulting from the simulations were corrupted with additive white Gaussian noise (SNR=30 dB) to mimic the unavoidable presence of the measurement noise.

#### 2.4.3. MWI Imaging Results

For each set of multi-frequency MWI raw data, two TSVD images were computed—one for the tumor and one for the spinal cord. To this end, the two operators, $L_{\Omega_{1}}$ and $L_{\Omega_{2}}$, were built from the knowledge of the total field inside the ROIs at the beginning of the treatment, $E_{tot}^{\left[0\right]}$, and of the morphology of the tumor and the spinal cord, respectively. Note that, for the sake of simplicity but without loss of generality, in this study, $\Omega_{1}$ and $\Omega_{2}$ were taken as square domains enclosing the anatomical site of interest. The whole imaging domain had $N_{x}=N_{y}=128$ pixel dimensions, from which the two ROIs were cropped around the pixels belonging to the tumor and the spinal cord, leaving $N_{x}=N_{y}=5$ and $N_{x}=N_{y}=6$, respectively. Finally, the TSVD images were computed using Equation (Equation 7) with $P_{cut}=1$ for both cases.

#### 2.4.4. CNN Implementation: Categorical Labels

For the considered temperature ranges, the ground truth classes in C were set as:(9)$$C_{1}\left(\Delta T\right)=\left\{\begin{array}{cc} 0 & \mathrm{if}\;\Delta T<3\;{}^{\circ}C\\ 1 & \mathrm{if}\;3\leq \Delta T<7\;{}^{\circ}C\\ 2 & \mathrm{if}\;7\leq \Delta T\;{}^{\circ}C \end{array}\right.$$
corresponding to {0,1,2} = {unheated, therapeutic, hot}. Note that the typical HT treatment therapeutic regime was T=[40,43] °C, i.e., ΔT=[3,6] °C.

Similarly, for the spinal cord,
(10)$$C_{2}\left(\Delta T\right)=\left\{\begin{array}{cc} 0 & \mathrm{if}\;\Delta T<2\;{}^{\circ}C\\ 1 & \mathrm{if}\;2\leq \Delta T\;{}^{\circ}C \end{array}\right.$$
with {0,1} = {unheated, hot}. Here, ΔT=1 °C was considered unheated for the purposes of these experiments. In clinical practice, a more stringent threshold could be used.

#### 2.4.5. CNN Implementation: Training

The necessary datasets for training the two CNNs were built by performing a number of simulations in the conditions described above. Each simulation provides an instance of the training set made by a pair δχ, $C_{c}$, i.e., the differential contrast estimated via TSVD in the relevant ROI and the corresponding categorical label.

In total, $N_{1}=3000$ samples for the tumor CNN and $N_{2}=2000$ samples for the spinal cord CNN were simulated. Each sample was built by randomly varying the EM properties of all tissues within a ±0.2-wide interval around their average values. For each tissue, this corresponds to modeling the real and imaginary parts of the complex dielectric permittivity as two normally distributed variables, $\varepsilon_{r}^{’}∼N\left(\varepsilon_{r}^{’},0.1^{2}\right)$ and $\varepsilon_{r}^{’’}∼N\left(\varepsilon_{r}^{’’},0.1^{2}\right)$. Modeling tissues with higher standard deviations than 0.1 is possible, but would possibly require increasing the number of samples $N_{1}$ and $N_{2}$ to obtain performances comparable to those reported in Section 3.

For implementation purposes, images had to be fed into the CNNs as real-valued quantities; therefore, the real and imaginary parts of the TSVD images had to be split and fed as stacks of images with $N_{chan}=2$ channels, such as $\delta \chi \in R^{N_{x}\times N_{y}\times N_{chan}}$. In Figure 4, two pairs of TSVD images (one for each CNN) are shown as examples of the input data.

The training was carried out using a K-fold cross-validation scheme [22], where some instances from the two datasets $N_{1}$ and $N_{2}$ were removed for each training iteration. In K-fold cross-validation, a dataset was split into $N_{fold}$ sub-datasets of equal sizes $N_{val}$ to carry out $N_{fold}$ training. In each iteration, one of the sub-datasets was removed from the dataset, leaving $N-N_{val}$ instances for training and $N_{val}$ instances for validation of the model’s performance. Once the $N_{fold}$ training was concluded, the performance could be assessed by averaging the results across folds. In this work, $N_{fold}=10$ was for both datasets (tumor, spinal cord). The validation split was $N_{val}=300$ for the CNN whose duty was to monitor the temperature status of the tumor, whereas $N_{val}=200$ for the CNN in charge of monitoring the temperature status of the spinal cord. The Adam optimizer [31] was used with a learning rate $\eta =10^{-4}$ and a batch size of $N_{b}=16$. The trainings were run for a maximum of $N_{t}=300$ epochs with an automatic stopping criterion of 10 consecutive epochs without improving the validation loss. Note that an epoch of *t* corresponds to a complete pass of all the instances in a given fold.

#### 2.4.6. Performance Assessment Metrics

Metrics to evaluate the performances of the classification problems involving categorical labels are based on the positive (P)/negative (N) count, such as the Dice similarity coefficient (DSC) [32]:(11)$$DSC=\frac{2\cdot TP}{2\cdot TP+FP+FN}$$
where TP, FP, and FN correspond to the true positive, false positive, and false negative count, respectively. More precisely, TP corresponds to the statistical count of correctly labeled instances into one of the temperature status categories. In contrast, FP and FN correspond to the mistakes incurred by the CNNs when labeling the instances. For example, a TP would be a variation of ΔT=4 °C, correctly labeled as therapeutic when considering the tumor CNN. Additionally, a second metric called the Matthews correlation coefficient (MCC) [33] was calculated as well:(12)$$MCC=\frac{TP\cdot TN-FP\cdot FN}{\sqrt{DM}}$$
where DM=(TP+FP)(TP+FN)(TN+FP)(TN+FN). As opposed to DSC, MCC employs a true negative count, TN, i.e., the statistical count of the correctly labeled instances as not belonging to a category. In general, MCC tends to show lower values than DSC and, therefore, provides a more conservative performance evaluation. Both metrics achieve their highest values at 1.00.

## 3. Results

After the CNN training is carried out on all the folds, the performance are assessed using the selected metrics (DSC and MCC). The metrics are computed class-wise on the validation split of each fold. Then, the final result for each class is obtained by averaging among all the obtained values. Table 1 and Table 2 report the metrics for the tumor CNN and for the spinal cord CNN, respectively.

## 4. Discussion

Concerning the tumor CNN, in agreement with the expectations, MCC tends to report lower values than DSC (see Table 1). Nevertheless, the achieved values are comparable between the two metrics for all classes. Moreover, there are no major differences in the performance of any specific class over the rest, only a slightly superior performance of the unheated class over the other two. More in detail, the overall number of misclassified samples is 26, which corresponds to about 8.5% of the whole validation split. In particular, for 2 samples (0.6%) the CNN predicted that the temperature was above the therapeutic one of 42 °C, while it was not. Of these, only 1 (0.3%) was above the therapeutic range. For another 15 samples (5.0%), the CNN predicted that the therapeutic regime was not reached, while instead it was. For 9 (3.0%) overheated samples, the CNN erroneously attributed the therapeutic label. These results are summarized in the confusion matrix reported in Figure 6.

For the spinal cord CNN, the performance scores are on average slightly worse than those of the previous case (see Table 2). This suggests that the task of classifying the spinal cord temperature status is more challenging, even though less classes were considered in this case. Such an outcome may be explained by two aspects. The first one is the further distance of the spinal cord from the antennas in the selected setup configuration (see Figure 4) as compared to the rather superficial location of the tumor. The second aspect is the smaller temperature range which is considered. Both these aspects result in scattered fields which are possibly more corrupted by noise and thus in less accurate TSVD images. Going more into detail, the reported metrics correspond to 16 misclassified samples, i.e., 8% of the validation set, evenly distributed between the two classes. Interestingly, 6 out of the 8 cases in which the spinal cord was erroneously labeled as unheated correspond to cases in which the temperature of the sample was 39 °C. From a practical perspective, this suggests that the network performs better when the spinal cord temperature is above 39 °C and thus that, even with some delay with respect to actual moment in which the hot spot occurs, it would eventually reveal it.

To figure out how to further improve the performance of the proposed framework, it is worth to analyze the behaviour of the categorical cross-entropy in the training and in the validation. Figure 7, reports the categorical cross-entropy of the fold whose performance was closest to the average in each dataset for the tumor CNN and the spinal cord one. Both plots confirm that the training is not affected by overfitting and hence that the adopted training is suitable to optimize the CNNs. In addition, the validation losses are on average always below the corresponding training losses across epochs, which is consistent. On the other hand, the cross-entropy for the validation splits exhibits an increasingly oscillating behavior with the epochs, especially for the spinal cord CNN. This suggests that the considered number of epochs is sufficient (if not redundant) and that there is room for performance improvements if the datasets get increased beyond $N_{1}=3000$ and $N_{2}=2000$.

## 5. Conclusions

The exploitation of a DL-MWI approach for HT monitoring was proposed and initially validated in silico for the case of neck tumors. The proposed approach is a physics-assisted DL-MWI framework, in which an image generated via MWI is input to a CNN that classifies it into one of several predefined categories. In particular, since tissue EM properties exhibit very small variations in the temperature range of HT, the ISP can be cast in terms of a linear inverse problem, whose solution can be reliably and efficiently built using the TSVD algorithm. In addition, this allows for achieving two separate images, i.e., one for the region of the tumor to be treated and one for the healthy tissue to be protected. Accordingly, two CNNs are exploited to provide a classification of hyperthermia measurements into labels reporting the temperature status in the different ROIs. The approach assumes prior knowledge of the morphology and tissue segmentation of the treated region, based on the fact that such information can be obtained from the treatment planning stage routinely performed in clinical HT.

The compelling results shown by the proposed approach on the validation samples provide a good basis to progress toward the experimental demonstration. To this end, a more realistic numerical scenario could be used to validate the approach, which includes representing realistic antennas, evaluating the temperature distribution using the bio-heat equation or other models proposed in the literature [34], taking into account additional information on the dependence of EM properties on the physiological changes of the tissue during the hyperthermia treatment, and extending the framework to the 3D geometry. Notably, all of these aspects only imply increased computational complexity but do not require changing the overall architecture of the approach. In addition, to better integrate the proposed approach in the clinical flow, an interesting development could be that of linking the classification labels to the usual HT monitoring parameter. For instance, the way in which the classification labels are designed has a strong connection with T90 since the therapeutic regime is only assigned when all of the pixels belonging to the tumor ROI reach the lower boundary of the relevant permittivity range.

Our final comment concerns the extent to which motion artifacts, which are critical for MR thermometry, impact the performance of the proposed framework. In MWI, inaccuracies introduced by movements can be modeled as additional sources of noise on the measured data. Since MW data acquisition can be performed very fast (faster than the breath rate), several measurements can be acquired and then averaged in such a way as to remove noise and achieve reliable imaging results. On the other hand, the movements of the ROI affect the co-registration of the treatment-planning image needed to implement the kernel of the inversion algorithm. However, MWI has a lower resolution than MR, which in this case turns into a sort of advantage as “small” enough motion shifts are basically not seen by the MWI device.

## Figures and Tables

**Figure 1 cancers-15-01717-f001:**
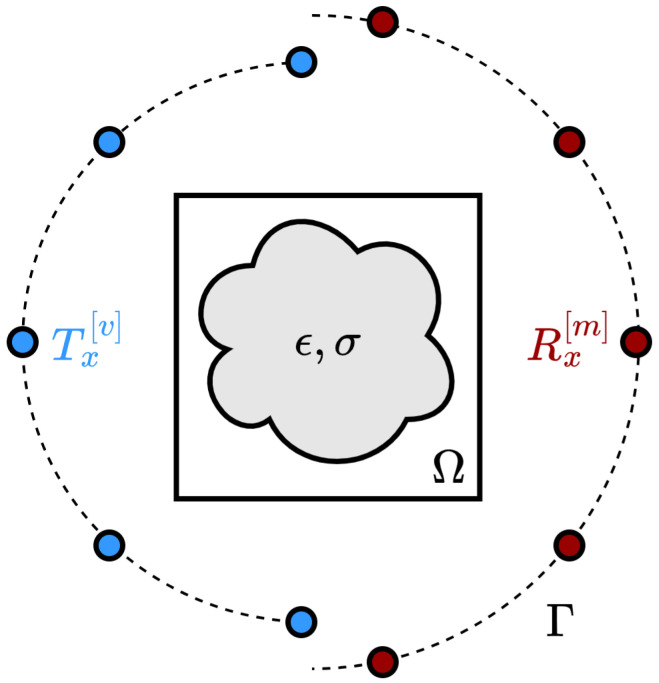
Typical MWI setup configuration. A number of transmitting $T_{x}^{\left[v\right]}$ and receiving $R_{x}^{\left[m\right]}$ antennas surround the imaging domain Ω.

**Figure 2 cancers-15-01717-f002:**
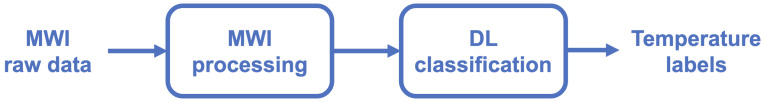
The basic ingredients of the proposed DL-MWI framework for temperature monitoring during HT treatments.

**Figure 3 cancers-15-01717-f003:**
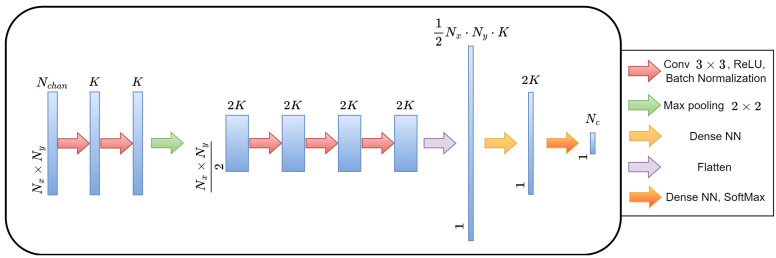
Convolutional neural network diagram for classification problems.

**Figure 4 cancers-15-01717-f004:**
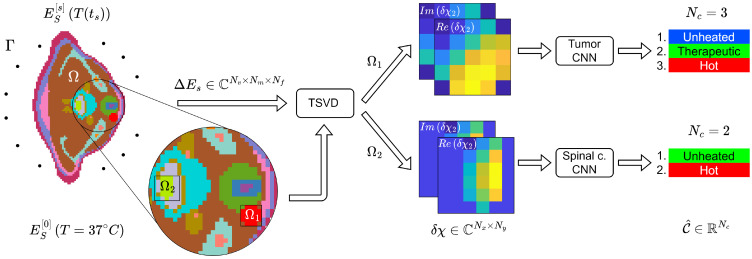
DL-MWI framework for monitoring temperature in neck tumor HT. Two separate CNNs simultaneously check the temperature status of the treated tumor and the spinal cord. In the figure, $\Omega_{1}$ denotes the tumor region and $\Omega_{2}$ is the spinal cord to be preserved.

**Figure 5 cancers-15-01717-f005:**
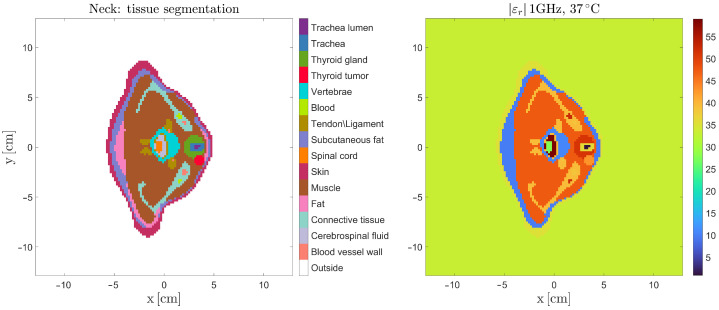
The phantom adopted for the numerical study. Tissue segmentation (**left**) and electromagnetic properties (**right**).

**Figure 6 cancers-15-01717-f006:**
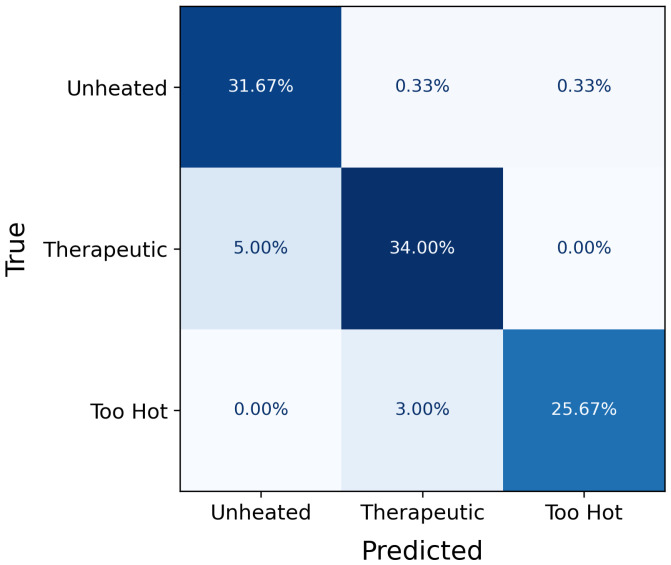
Confusion matrix for the validation samples of the tumor CNN.

**Figure 7 cancers-15-01717-f007:**
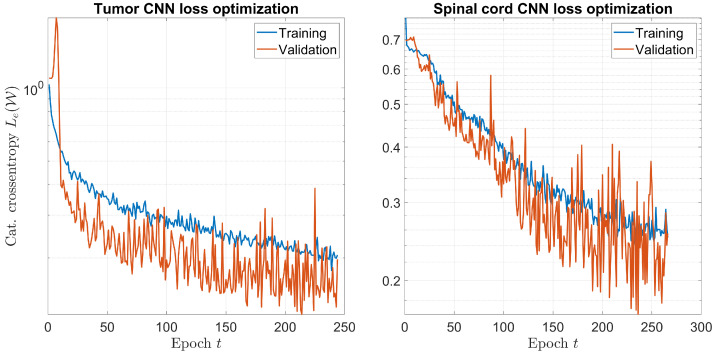
Categorical cross-entropy optimization of the considered CNNs versus the number of epochs.

**Table 1 cancers-15-01717-t001:** DL-MWI Tumor CNN performance assessment.

Class	*DSC*	*MCC*
Unheated	0.953	0.928
Therapeutic	0.932	0.887
Hot	0.912	0.878

**Table 2 cancers-15-01717-t002:** DL-MWI Spinal cord CNN performance assessment.

Class	*DSC*	*MCC*
Unheated	0.920	0.855
Hot	0.907	0.855

## Data Availability

The data generated for the numerical simulations presented in this study are available on request from the corresponding author.

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
