# Peer review of "Hyperthermia Treatment Monitoring via Deep Learning Enhanced Microwave Imaging: A Numerical Assessment"

_cancers, 2023, doi:10.3390/cancers15061717_

Round 1
Reviewer 1 Report
The manuscript presents an interesting application of Deep Learning tools to hyperthermia temperature monitoring. A minor revision is suggested.
Authors are invited to discuss if the inaccuracies introduced by movements in the HT-MR technique (see lines 39-40) are also present, or not, in the proposed MWI-DL approach. In other words, is the statement of lines 81-83 applicable, considering that some movements (e.g. breathing, bowels, blood vessels) are inevitable?
Referring to lines 304-308, could the authors say what appens if real and imaginary parts of the complex permittivity are assumed with a larger variance?
On line 125 I suppose that Nabla operator is a vectorial one, so the expression in brackets lacks unit vectors.
On line 160, where ROI is used for the first time, I suggest to specify it as Region of Interest.
Author Response
The manuscript presents an interesting application of Deep Learning tools to hyperthermia temperature monitoring. A minor revision is suggested.
Thank you for appreciating our work, point-to-point reply to your comments follows.
Authors are invited to discuss if the inaccuracies introduced by movements in the HT-MR technique (see lines 39-40) are also present, or not, in the proposed MWI-DL approach. In other words, is the statement of lines 81-83 applicable, considering that some movements (e.g. breathing, bowels, blood vessels) are inevitable?
Thanks for this comment. Inaccuracies introduced by movements are of course affecting MWI measurements as well. However, these are expected to be less impacting with respect ot HT-MR case. In fact, while MR imaging is very sensitive to the position of the imaging domain, MWI builds the image based on the field measured outside of the body. As such, movement is encoded as an additional source of noise on the data. Now, considering that MW data acquisition is very fast (faster than breath rate), several measurements can be acquired and then averaged to clean the data and perform the imaging safely. In addition, due to the much lower spatial resolution of MWI as compared to MRI, “small” motion shifts can be not “seen” by the MWI device. A comment on this limitation was added to the conclusions section.
Referring to lines 304-308, could the authors say what happens if real and imaginary parts of the complex permittivity are assumed with a larger variance?
The variance affects the accuracy of the classifier predictions. As such, if a large variance is expected, possibly a larger number of training samples N has to be considered according to enlarged range of variability. We added a comment on this in the corresponding section.
On line 125 I suppose that Nabla operator is a vectorial one, so the expression in brackets lacks unit vectors.
Thanks for spotting the typo, we added the unit vectors to the Nabla operator.
On line 160, where ROI is used for the first time, I suggest to specify it as Region of Interest.
Thanks for this comment. We have added the needed specification.
Reviewer 2 Report
The authors suggest a new method of non-invasive temperature measurement in hyperthermia. In this article, they show the principles of the method and the theoretical feasibility in a simplified scenario.
So far, it is not clear whether this procedure lead to clinical application. At this state of the development, there are many simplifications (only one slice, point antenna, small ROIs etc.), approximations and uncertainties included. It is also questionable whether the difficulties of mathematical and computing effort are solvable for realistic systems.
However, in the case of success the benefit of non-invasive 3d temperature measurement in hyperthermia would be so high that it is worth to publish this article and continue the investigation of this method.
There are two critical questions the authors have to discuss in more detail:
First, is the classification procedure necessary? The basic idea of the article is that the temperature influences the EM properties of the tissue and therefore, the temperature can be calculated by these changes. If this is the case, it would be possible to get directly the temperature distribution from the solution of the MWI. Then, the temperature distribution is sufficient to examine the HT treatment and calculate quality parameters (e.g. T90, Tmean and CEM43°) and no further classification procedure is necessary. Furthermore, this temperature distribution would include more information than the simply classification to improve an insufficient temperature distribution by antenna steering. Therefore, the authors have to clarify, what the advantage of the additional classification procedure is.
Second, is there a unique relation between EM properties and temperature? A change of the EM properties could caused not only by a temperature change. Maybe also during the treatment caused by physiological changes (e.g. perfusion). A separation of the different causes could be difficult.
Additional, the relation between the EM properties and the temperature could be different in different persons and situations. Therefore, is it really possible to get the real temperature distribution from the electric parameter distribution in an unique way? The authors have to discuss this point.
Author Response
The authors suggest a new method of non-invasive temperature measurement in hyperthermia. In this article, they show the principles of the method and the theoretical feasibility in a simplified scenario. So far, it is not clear whether this procedure lead to clinical application. At this state of the development, there are many simplifications (only one slice, point antenna, small ROIs etc.), approximations and uncertainties included. It is also questionable whether the difficulties of mathematical and computing effort are solvable for realistic systems. However, in the case of success the benefit of non-invasive 3d temperature measurement in hyperthermia would be so high that it is worth to publish this article and continue the investigation of this method.
Thank you for appreciating the essence of our contribution. We are aware the way to the application is still long, but we trust that this initial feasibility results suggest it is worth to purse the research effort needed to perform a more meaningful validation.
In doing so, it can be noted that the adopted assumptions have an influence only the dataset used for the training of the algorithm. Adding more complexity (or making less assumptions) to move towards realistic systems would require longer simulation times, but the methodology would remain almost identical. With respect to the computational burden needed to simulate more realistic scenarios, efforts are being made to cut down simulation time, so that it could became affordable possibly even within the treatment planning window.
Below we report reply to your very meaningful comments.
There are two critical questions the authors have to discuss in more detail: First, is the classification procedure necessary? The basic idea of the article is that the temperature influences the EM properties of the tissue and therefore, the temperature can be calculated by these changes. If this is the case, it would be possible to get directly the temperature distribution from the solution of the MWI. Then, the temperature distribution is sufficient to examine the HT treatment and calculate quality parameters (e.g., T90, Tmean and CEM43°) and no further classification procedure is necessary. Furthermore, this temperature distribution would include more information than the simply classification to improve an insufficient temperature distribution by antenna steering. Therefore, the authors have to clarify, what the advantage of the additional classification procedure is.
Thank you very much for your comment. The idea behind the paper was to provide an initial demonstration of the fact that we can cast the MWI-HT monitoring problem without explicitly providing an estimation of the temperature but rather giving the clinician immediate information on the ongoing treatment in some critical sites. The reason for this is that, although changes in temperature are embedded in the EM properties, such changes are not easily interpretable from the images coming from the MWI system. In this respect, while a Deep Learning algorithm could be used to extract the temperature distribution from the information in the MWI images, this approach would require a significant effort in training the network, as this would need to be capable to perform a very complex regression task and provide a quantitative output. Conversely, the approach proposed in this paper was to go cast the problem in terms of a classification task in order to lighten the complexity of the task the network has to learn. In the proposed approach, the goal of the DL algorithm is to provide a qualitative information which can help the physician in spotting potential hazards.
Regarding the reviewer’s comment on the use of quality parameters like T90. Note that the way in which the classification problem was designed has a strong connection to T90 since the therapeutic regime is only retrieved when all the pixels belonging to the tumor reach the lower boundary of this range. Accordingly in the conclusion we mentioned that a possible development of the present work is to extend the classification labels to the typical HT parameters.
Second, is there a unique relation between EM properties and temperature? A change of the EM properties could be caused not only by a temperature change. Maybe also during the treatment caused by physiological changes (e.g., perfusion). A separation of the different causes could be difficult. Additional, the relation between the EM properties and the temperature could be different in different persons and situations. Therefore, is it really possible to get the real temperature distribution from the electric parameter distribution in a unique way? The authors have to discuss this point.
Thank you for this insightful comment. We agree that during hyperthermia treatments the increasing temperature changes e.g., the tissues’ blood perfusion and this can in turn change the dielectric properties of the tissue. To what extent these changes (blood perfusion, etc) influence the dielectric properties is not known at present. In fact, in our knowledge, dielectric properties of tissues have never been measured in vivo during a hyperthermia treatment. Accordingly, the data used to model the temperature dependence were taken from ex vivo studies (ref. [9] of the manuscript). While this point evidences a clear gap in knowledge, once such measurements are performed, the achieved relationship between dielectric properties and temperature can be readily implemented in the proposed DL-MWI architecture. A comment on this was added to the conclusions section.
The inter-subject variability is an aspect to be carefully undertaken. In this respect, the data-driven nature of the proposed approach can be helpful, as acquiring data for many controlled cases can be exploited to train a classifier which learns how to handle variability. In addition to this, use of data acquired at multiple frequencies can be exploited to increase data-diversity and possibly counteract inter-subject variability.
Some of the above comments have been included in the conclusions of the paper.